# A Conceptual Framework Proposal for a Noise Modelling Service for Drones in U-Space Architecture

**DOI:** 10.3390/ijerph19010223

**Published:** 2021-12-25

**Authors:** Tommy Langen, Vimala Nunavath, Ole Henrik Dahle

**Affiliations:** 1Department of Science and Industry Systems, University of South-Eastern Norway, 3616 Kongsberg, Norway; tommy.langen@usn.no (T.L.); vimala.nunavath@usn.no (V.N.); 2Indra Navia AS, 1383 Asker, Norway

**Keywords:** noise model, Unmanned Aircraft System Traffic Management, drones, UAVs, accepted noise levels, U-space architecture

## Abstract

In recent years, there has been a rapid growth in the development and usage of flying drones due to their diverse capabilities worldwide. Public and private sectors will actively use drone technology in the logistics of goods and transporting passengers in the future. There are concerns regarding privacy and noise exposure in and around the rural and urban environment with the rapid expansion. Further, drone noise could affect human health. European Union has defined a service-orientated architecture to provide air traffic management for drones, called U-space. However, it lacks a noise modelling service (NMS). This paper proposes a conceptual framework for such a noise modelling service for drones with a use case scenario and verification method. The framework is conceptualized based on noise modelling from the aviation sector. The NMS can be used to model the noise to understand the accepted drone noise levels in different scenarios and take measures needed to reduce the noise impact on the community.

## 1. Introduction

Unmanned aerial vehicles (UAVs), also known as drones, have been exemplified as the most disruptive technology in human history and experts predict that by the year 2030 there will be a billion drones in the world [1]. Thus it is regarded as one of the most rapidly developing aspects of aerospace science and technology [2]. According to a report from the Federal Aviation Administration (FAA), it is estimated that the fleet of drones will be more than doubled from 1.1 million vehicles in 2017 to 2.4 million units by 2022 [3].

The drone technology has been primarily used earlier for military purposes. However, at present, it is broadly used in many domains for several purposes such as investigation of crops, observation of weather, relay broadcasting and communication, investigation of the extent of damage during disasters, recognition of traffic flow, and unmanned security. In particular, several global IT companies such as Amazon, Google, and Facebook have announced that they will actively use drone technology in logistics. It is expected that the applications of drone technology in the private sector will rapidly expand in the near future [4]. Other companies such as Volocopter, Uber, and Hyundai are working on drones to use them as unmanned taxis for transporting passengers in future [5,6].

Even though there are many advantages of using drones, they have also raised several concerns regarding public safety, security, privacy and community noise exposure as they are flown in and around the rural and urban environment. With this, they have the potential to expose large portions of communities to a new noise source [3,7]. Here, noise is often described as an unwanted sound and is known to have several adverse effects on human health due to various noise levels produced by different drone types [8,9,10].

European Union has defined the U-space blueprint [11] to enable both development of the drone industry in Europe and ensure safe and efficient drone operations [12]. The U-space blueprint presents a service-orientated architecture to provide air traffic management (ATM) to drones. It is described in detail in the SESAR roadmap for the safe integration of drones into all classes of airspace [13,14]. U-space contains several services, and these services aim to promote the development of smart, automated, interoperable, and sustainable traffic management solutions. These will be a key enabler for achieving a high level of integration. The most critical success factor for U-space operations will be the ability to identify solutions that allow drones and all other airspace users (unmanned and manned) to operate safely, securely, sustainably, and efficiently in a managed and fully integrated airspace by preventing an undue impact on operations currently managed by ATM [15].

Even though many services are incorporated in the U-space architecture, a service related to noise modelling is missing. It is essential to introduce noise modelling into U-space because many studies show that several adverse effects on human health come from noise exposure. Such as hearing impairments and tinnitus [16], cardiovascular and metabolic repercussions [17], learning impairment [18], sleep disorders [19], and annoyance [10,20]. As a dramatic consequence, the world health organization estimates that noise effects on health may be responsible for up to 1.6 million disability-adjusted life-years (i.e., the potential years of life lost due to premature death) in Western Europe [21]. One of the noise sources can be related to transportation (aircraft, railway and road) which contribute to both the perception of noise disturbance and health impact due to several factors, such as long-term exposure [22], noise re-occurrence [23] or low-frequency noise [24]. However, it should be noted that, to a lesser extent, noise annoyance and some noise effects may come from other sound sources, such as community noise [25], industrial noise [26] or recreational and leisure noise [27]. Lastly, the total annoyance can also be the result of combined noise, such as between road traffic [28] and industrial noise sources [29].

In response to these significant health and societal challenges, the European Union implemented the Directive 2002/49/EC [30] in 2002, which is related to the assessment and management of environmental noise. The goal is to define an EU common approach to avoid, prevent or reduce the harmful effects due to environmental noise exposure, mainly due to transportation and industrial sources. Based on all the data acquired through the application of this directive, in 2012, almost 122 million inhabitants were subject to noise levels above the limit of 55 dB(A), primarily from road traffic and railways. This limit is the threshold to consider the health effects of noise on inhabitants, including 14 million at extremely annoying levels [31]. Due to the serious nature of the possible adverse effects of noise exposure, it is important to understand and model drone noise. The International Civil Aviation Organization (ICAO) has discussed and declared that there are possibilities of various problems in the future which can be generated from drone noise. They state, “as new products and aircraft come into use, it may become apparent that additional noise and emission standards are necessary” [9,32]. This justifies why studies are still needed to model the noise and reduce noise pollution, whether in road, rail, air traffic or industrial activities. For recent studies on this, we refer the reader to [29,33,34,35].

In general, noise metrics are used to quantify the community response to various noise exposure levels. Most of the existing noise estimation approaches are derived primarily for manned civil aviation aircraft which are based on well-researched source noise and sound propagation and transmission loss models for those aircraft [8]. Such detailed information is minimal for expected future drones and a key knowledge gap for UAVs. Thus, as there is a lack of noise modelling service in the U-space architecture, the objective of this research is to propose a conceptual noise modelling service framework for drones to model the noise in order to understand the accepted noise levels in different scenarios. In order to achieve the objective, it is essential to understand the theory related to noise, noise models and metrics and then review the literature to find different drone types, the important elements used for modelling the noise, and the acceptable noise levels in different environments to quantify the distinct impact of large-scale unmanned traffic operations.

The rest of the paper is organized as follows. The existing literature on noise modelling is presented in Section 2. Section 3 explains the research methodology used to propose the conceptual framework. Section 4 presents our proposed conceptual framework for noise modelling service with a use-case scenario and verification methods to show how our proposed noise modelling service can be utilized in real life. Finally, we sum up our paper in Section 5 with conclusions and possible future work.

## 2. Literature on Drone Noise Management

Apart from developing the UAVs themselves, the use of drones in various fields has been studied. Further, much research has been done in the commercial aviation industry to model the noise, to find solutions for community annoyance, and to develop air traffic systems. However, research on noise modelling and management for drones are scarce. Hence, we present the existing literature on drone noise, noise reduction, community annoyance, and noise modelling using various methods in the following paragraphs.

Ryan et al. [7] investigated the noise which was produced by contra-rotating UAV propellers during hovering. This was done by testing various configurations of contra-rotating UAV propellers mounted statically in an anechoic chamber. They also studied the effects of propeller diameter by testing different configurations of 12 and 15 inches propellers. In addition, they also investigated the effects of propeller spacing, blade number and rotational speed and. Their experiments observed that interaction tones were a dominant contributor to the overall noise level, which also contained rotor-alone tones and broadband noise.

John et al. [9] examined the noise emitted by drones to mitigate the noise by determining the suitable operating conditions and flight paths. In addition, for modelling drone noise, the authors also used and studied the suitability of the software package ‘iNoise’. From their experimental results, the authors found out that if the drones are flown at high altitudes, high speeds, vertical takeoffs and landings, the noise emission of the drones could be mitigated. The measured sound power level was actually recorded from a small quadcopter, they assume that larger commercial drones would have a higher sound power level. So, they suggest further investigations should be done for noise mitigation measures. Further, the iNoise software used in the research demonstrated the suitability for modelling noise emissions.

Jeon et al. [36] investigated use of a deep neural network to detect commercial hobby drones in real-life environments. They analyzed sound data to contribute to a system for detecting drones used for terrorism. In order to achieve the objective, the authors presented a method for binary classification to detect the commercial hobby drones based on the sound event detection. They also utilized classification methods, i.e., a Gaussian Mixture Model (GMM), Convolutional Neural Network (CNN), and Recurrent Neural Network (RNN), for drone sound detection. Their empirical results show that RNN models showed the best detection performance with a F1-Score of 0.8009 by using a 240 ms of input audio with a short processing time. The authors claim that this indicate their applicability to real-time detection systems.

Chun et al. [37] proposed the noise reduction method using a deep convolutional denoising autoencoder for eliminating drone flying and wind noises. To do this task, the authors have generated a training dataset by mixing drone flying and wind noises in clean speech signal. Then the authors trained the convolutional neural networks. Their experimental results show that the trained model could reduce both the drone flying and wind noises.

Minghuan et al. [38] proposed a deep learning modulation classification method in non-Gaussian environment for noise detection. Their experimental results show that the proposed algorithm can effectively constrain the sharp pulse in non-Gaussian noise and improve the modulation recognition accuracy. In addition, the experiments show that their proposed method is robust to the noise characteristic exponent.

Carvalho et al. [39] developed a noise model to study the impacts of the environmental noise on motorway A23 in the stretch between Castelo Branco–Alcains (Portugal) and modelled the noise variation for establishing mitigation measures. For measuring the road traffic noise, the authors used French calculation method NMPB-Routes-96 (French standard XPS 31-133) by using several indicators. The results from the experiments show that the modelled noise can be a valuable tool for monitoring noise impact in the surroundings of motorways.

Mark et al. [40] explored the existing regulatory framework in Australia, highlighting the suitable noise certification standards and noise metrics. The results of their research are stated as “the current ICAO noise certification standards may provide useful guidance for urban air mobility (UAM) noise certification, however they are not directly applicable. Likewise, maximum sound power levels within the current European standards for Remotely Piloted Aircraft Systems (RPAS) provide a useful guide, but may not directly apply, particularly for RPAS delivery systems operating in an urban environment”. The authors performed a hypothetical UAM noise model test case. The results showed that it was possible to generate typical aircraft noise metrics. They discovered some limitations with the software they used, such as the limited ability to accommodate noise reflections from building facades in an urban central business district environment.

Ciaburro et al. [41] reported a methodology for measuring the noise emitted by small UAVs during flight inside a closed environment. Acoustic measurements of the noise emitted by a drone inside a large environment (12.0 × 30.0 × 12.0 m) were performed. The noise was measured with a sound level meter placed at different distances (5, 10, and 15 m), to characterize the noise in the absence of anthropic noise. They found that it was possible to distinguish typical UAV noise at the frequency of 5000 Hz, despite being in an environment that simulated buzz of people and background music. The authors concluded that there is possible to discover the presence of small UAV’s using acoustic measurement.

Baron et al. [42] determined a noise source using array measurements and then developed a supervised learning model. Their acoustic array allowed to establish where the source direction of arrival. While the spatial filtering was used to improve the signal to noise ratio. With a gained focused signal, they could use it to characterise the source. They characterized the signals to classify sources as drone or some other noise, based on two learning models. The first used a classic Support Vector Machine model, using two classes of drones and noise. The second was built upon a One Class Support Vector Machine algorithm, where only the drone class was learned. The experimental results gave an error on elevation angle bounded to 3.7° on localization. While the identification results gave 99.5% accuracies for the two classes approach, and 95.6% for the one class approach.

Antonio et al. [43] explored the effects of drone noise in seven different types of urban soundscapes. The urban soundscapes are highly impacted by road traffic noise. So, if a drone is flown in the urban soundspace, the presence of drone noise leads to small changes in the perceived loudness, annoyance and pleasantness. From the experiments, the researchers noticed that with a reduced road traffic noise in soundscapes, the drone noise was significantly higher perceived loudness and annoyance and lower pleasantness. The results show that there was a increase in annoyance from 2.3 ± 0.8 (without drone noise) to 6.8 ± 0.3 (with drone noise).

Kang et al. [44] proposed a software platform for networked UAVs to solve the noise problem and developed an application to group display. To measure the service quality of the proposed platform, the authors developed a small-scale test-bed and performed several experiments. The experimental results showed that from the proposed noise reduction, they could attain a speech signal with up to 67.5% same as the original signal. This means that the proposed noise reduction method could outperform active noise control and spectral subtraction with an approximation of 53.1% and 39.6%, respectively.

Kloet et al. [45] measured the noise generated by a single typical multi-rotor unmanned aircraft system (UAS) arm which includes propeller and support structure. During their experiments, the researchers measured the noise and found out that it was because of the noise from the the propeller flow field and its interaction with the airframe support structure (arm). They also investigated to understand how the effect of the flow field affect a typical airframe support structure. From the experiments, the researchers propose to minimize the noise which was generated aeroacoustically and recommend to reduce the additional noise.

Cabell et al. [46] investigated and measured the noise from small UAVs. The authors considered four UAVs including an internal combustion-powered model airplane and three battery-powered multicopters. The small UAVs were flown over flyover and reading were collected. Based on the collected reading, the authors discussed the basic noise characteristics such as spectral properties, sound levels metrics such as sound pressure level, effective perceived noise level, and sound exposure level. From their experimental results, the authors found out that due to the size and aerodynamic characteristics of the multicopters, the flight path of the UAVs was affected by atmospheric disturbances. Due to this unsteadiness, the authors explored the spectral variations in both hover and flyover conditions for the UAVs. Because of the time-varying noise, the authors claim by using using conventional sound level metrics, it becomes difficult to predict the human annoyance due to the UAVs’ noise.

Bulusu et al. [47] examined to provide the estimation for UAVs’ ambient noise levels which are flown in low-altitude uncontrolled urban airspace. To achieve the objective, the researchers first flew the UAV over urban areas and simulated the unmanned traffic to estimate the noise footprint which was generated over a day. During the simulations, four metrics namely Leq, Ln, A55 and P55 were computed. From the simulations, the researchers captured density of the increasing traffic, noise source and different noise levels on varying operational altitudes. From the simulated results, the authors mention that when UAVs are flown above 200ft altitude of high-speed, then the noise levels genereted by the UAVs will not be any hassle for the community.

Oeckel et al. [48] introduced several validation approaches to reduce the UAV noise. The performed experiments are related to precisely observe the natural blade oscillation by using a high-speed camera and also wanted to inspect the dynamic unbalances of the propeller blades. From the experiments, the authors first identified the falsifying impacts and then evaluated by using additional measurements such as noise levels, sound power, and source localization by using the acoustic camera. The experimental results proved that the achieved optoacoustic examinations were comparable and can be utilized further to reproduce the results.

Moshkov et al. [49] studied the effect of the UAV noise masking by the ambient noise level and presented the results of the acoustic tests of the Ptero-G0 UAV with a piston engine with level flight conditions. To perform the experiments, the data on the spectral portrait were obtained by the researchers. From their experiments, the researchers could determine that the first two tones of propeller noise in overall community noise level of UAV noise, and the piston engine noise audibility of the UAV. They also observed that when the flight speed got decreased approximately 30 km per hour, it led to decrease in the safe distance by approximately 370 m. When there is an increase in wind speed from 4 to 6 m/s and as a consequence of ambient noise levels, the UAV could perform light level at an altitude half as much, without any possibility of detection by the observer than at a wind speed of 1 to 2 m/s.

Christian et al. [50] studied to understand how the much humans are annoyed with the noise generated by the small UAV. For that, the researchers first did some field test and collected the sound recordings. By utilizing these recordings, psychoacoustic test was formulated and then executed. The experimental results showed that, from the collected data, they could see that there was a lack of parity between the noise of the recorded small UAV and a set of road vehicles. This might be because noise of the road vehicles were also got recorded in the data and this data was also included in the test and got measured by a set of contemporary noise metrics.

Although various scholars have worked on noise reduction, propeller design to reduce the noise of the drones, psychoacoustic effects on human-subjects due to noise, noise sources of drones, types of propellers and drones identification in open and closed spaces, sound detection and prediction, none of the above studies worked on drone noise modelling. Except Jeon et al. [36] whose work is not comparable to this work, as their focus was more on how drones should be flown to minimize noise impact, but not about adding a noise model to the existing U-space architecture. The other studies did not suggest what elements are important to consider to model the noise for drones. With this research, we add new scientific knowledge to the existing literature by proposing a conceptual framework for noise modelling for drones.

## 3. Materials and Methods

This study aims at investigating, if there are any existing models for drone noise modelling proposed in the literature. If no such models exist, we will propose a conceptual framework for a drone noise modelling service in the U-space architecture. In order to achieve this objective, we followed a research methodology proposed in [51] for the development of a conceptual framework, which can be seen in Figure 1. To propose a conceptual framework, three sources need to be considered: Experience, literature and theory/requirements. The sources for a conceptual framework are considered as principal elements which can be used as the basis for developing any framework.

After reviewing the literature and existing reports on drones, we found out that there is a digital infrastructure called U-space which has been proposed to support safe and efficient access to European airspace for millions of drones shown in Figure 2. It contains a set of decentralized services and procedures designed to support safe, efficient and secure airspace access for multiple drone (UAS) operations. The U-space shall be capable of ensuring the smooth operation of all categories of drones, all types of missions and all drone users in all operating environments. It provides the framework for routine drone operations and an interface to manned aviation management. The services in this architecture are organized, coordinated and managed by a federated (decentralized) set of actors in a distributed network of highly automated systems. For more details about involved actors in U-space architecture, we refer the reader to [15]. A set of standardized application programming interfaces (APIs) is used to exchange information among different services [15]. The U-space architecture consists of three main parts - here called environments. The first one is the authority mandate environment, the second one is the competent authorities environment, and the third one is the competitive industry environment. In the rest of the section, we only describe the authority mandate environment and competent authorities environment as these are of our interest.

One of the services of the authority mandate environment is Flight Information Management System (FIMS). It acts as a gateway to allow the data exchange between the U-space participants and ANSP systems. The Air Navigation Service Provider (ANSP) is responsible for providing directives to UAS operators via the U-space system. Further, they are also in charge of making relevant airspace information available to drone operators via the U-space system. The FIMS also provides a mechanism for shared situational awareness among all U-space participants and is a central component of the overall U-space ecosystem [15].

In Competent Authorities Environment (CAE) (see bottom left in Figure 2), the term USP refers to U-space Service Provider. The role of a USP is to provide access to all U-space services to drone operators, to pilots and/or to drones, or other operators visiting non-controlled very-low-level airspace [15]. The supplementary data service provider (SDSP) provides access to supplemental data like terrain, weather, and cellular coverage. Authority USP in CAE contains several tools and platforms to provide different services such as airspace management, registry search, safety services and so on. However, there is no tool or service available for modelling/management of the drone noise in the U-space architecture. In this research, we propose a new service called noise modelling which can be used for modelling the noise of drones to understand and reduce the noise impact on the community. Since the service should apply to the entire airspace covered by U-space and work across all USPs, it belongs in the competent authorities environment part of the U-space architecture. We show in Figure 2 where the NMS would fit in the U-Space architecture.

To propose the NMS framework, we also looked into understanding the drone noise and noise modelling theory and then found only one set of requirements for a noise modelling service. These requirements have been published in the Request for Proposal document by Airservices Australia for a Flight Information Management System (FIMS) which is part of a UTM (Unmanned Aircraft System Traffic Management) system for Australia [52]. This document outlines a Noise modelling Service (NMS) that uses a database of drone performance, planned and actual flight paths to calculate the noise output by each drone flight. The service shall generate cumulative noise maps (CNM) and notify other services in the UTM system, if noise limits are exceeded. Based on these requirements, we propose a conceptual framework for drone noise modelling. Below we have listed the most relevant requirements for eliciting the proposed NMS [52]:The noise management service shall incorporate a library of acoustic parameters for different UAV types capturing, at minimum, nominal dB output and nominal frequency range for standard operations.The noise management service shall use a free field acoustic propagation model and the UAV acoustic parameters to calculate the A-weighted sound level (dBA) at ground level along with the centre line of the Operational Intent Volume.The noise management service shall incorporate a GeoJSON Noise Threshold Map (NTM).The NTM will establish a basic grid over a geographic area and assign a maximum number of operations within different dBA level bands across different times of the day.The noise management service shall consolidate all operations into a dynamic Cumulative Noise Map (CNM).The noise management service shall determine actual noise impact (ANI) through the combination of UAV track data supplied by UAS Service Suppliers, the library of acoustic parameters for different UAV types, the free field acoustic propagation model, and capture of ANI in the CNM.When the CNM exceeds the limit specified by the NTM for particular regions, the noise management service shall generate a constraint through the constraint management service for the applicable time period.

After investigating the literature on drone noise modelling and government reports regarding noise models for commercial airports, we identified that certain elements are essential for modelling drone noise. Based on the experience, investigation and elements found from literature and theory/requirements, a conceptual framework for the noise modelling service for drones in the U-space architecture has been proposed and can be seen in Figure 2. The proposed framework is intended to cover the requirements listed above and fit into the U-space architecture [15], and is inspired by the European Civil Aviation Conference (ECAC) report on method of computing noise contours around civil airports [53]. The framework acts as a service in U-space, and contains three main components. Those are Noise modelling service (NMS), drone and environment data, and post-processor. The explanation of these components is given in detail in the below sub-sections.

## 4. A Conceptual Framework for Noise Modelling Service for Drones in U-Space

In this section, we present and describe the proposed conceptual framework which is shown in Figure 2. The proposed noise modelling framework aims to perform a calculation on a specific drone with the environment that will operate in, to achieve a theoretical sound level at a given point. We also describe the elements and their purpose and dependencies to other components. This section goes deeper into explaining the various data, an overview of the databases, a discussion of data quality, and insight into the post-processing.

### 4.1. Functional Overview of the Proposed Framework

The working procedure of the proposed framework is described below. First, U-space or a UTM system sends a request to the noise modelling service for using noise mitigation services related to a particular drone in a specific setting and environment. This could be to forecast or monitor an airspace. Then the NMS assesses the petition and initiates its three main processes. The pre-processor gathers and manages the necessary data, and then delivers key parameters to the noise calculation. The data must cover the drone and the environment it operates in. The data the pre-processor needs could be from various sources. In a U-space architecture, the data would be provided from Supplementary Data Service Providers, as seen in Figure 2.

To get a systematic overview, we suggest dividing the data into four categories: Drone database, drone flight profile, drone operation, and environment data. The noise emission profile uses related data and calculates a neutral and environmental free sound profile. The noise engine in the noise modelling service is the central process of the model and does the calculations of the sound emission and propagation. The process ends up with a maximum level of noise in a given period. If a drone movement is considered, it will simulate the noise over time, with a propagation path calculation. Then U-space receives the noise metrics in order to assess the situation and make necessary changes to the airspace. As the noise modelling service considers one drone in a given situation, post-processing is needed. Later these results should be combined with other elements. Post-processing can be utilized to predict noise in an airspace with multiple drones, to produce cumulative noise maps to check if a zone is within the governmental noise limitation regulations, or to study annoyance in the population.

### 4.2. Input Data and Databases

In the proposed framework, we need data to compute a theoretical noise value. The more comprehensive the data is, the more accurate the result will be. However, it can also be more complex. In this section, we present the types of data and databases needed in the noise modelling framework, as shown in Figure 3, and discuss the possibility of simplifying through categorizing drone types. Nevertheless, there might be concerns and limitations related to data quality.

#### 4.2.1. Drone Database

In the process of developing the framework, we aimed to simplify the process by drastically reducing the amount of various data to be compiled from numerous sources. As there are numerous drone models on the market and more to come, we tried to sort these into categories. Various product data can be used for categorization, such as weight, operational altitude, speed, number of rotors, fixed wings or engine power. If we were to organize the groups into all types of various classifications, the purpose of categorization would be eliminated due to the sheer number of groups.

We have chosen to focus on the European regulations and classifications of drones due to laboratory location. The EU uses EU-Regulations 2019/947, 2020/1058 and 2019/945 framework. The European Union Aviation Safety Agency (EASA) divides civil drone operations into open, specific, and certified categories. The EU-Regulations has introduced maximum noise levels, but only for class C1 and C2 so far [54,55]. Therefore, we have chosen to investigate, if the noise level can be distinguished based on EASA class identifications.

Drones will be marked with drone class within 2023, ranging from C0 to C6, primarily based on their Maximum Takeoff Mass (MTOM) and speed [54,55,56]. (C5 and C6 are C3-type drones with extra features for identification and safety, for use in the specific category.) As the regulations are continuously developed, additional C-classes might be added later.

The noise level from each drone class is presented in Table 1. One could assume that a heavier drone will always produce more noise due to an increase in needed power to fly. But as we can see from the data in the table, one cannot simply divide these classes according to their physical characteristics to achieve a clear noise level range. This advocates that the use of existing drone classes in EU alone, is not sufficient to categorize drone noise. Further research is needed into the possibility of how to group drone classes for easier database handling. If drone class grouping is not realistic, a suggestion is made that all commercial drones shall have in their product data noise level measurement. In the same fashion, as airplanes today need to undergo an aircraft noise certification program before being commercially used.

Due to the current limitations of using drone classifications for noise predictions, we advocate the use of a drone database with data for each drone model. The database contains mainly the performance characteristics and the acoustic properties of the drone. The performance data describes the physical characteristics, which can be used to analyse the forces and moments acting upon the drone. Valuable data could be the type of drone, engine, power, size, mass, speed, and number of rotors, if applicable. The manufacturer of the drones can provide these data. The acoustic data describes the sound emitted by the drone. The level of complexity dictates how much data is required. A basic level of acoustic data could be measurements of a predefined drone classification in a fixed atmospheric condition with a given distance. However, as stated in this section, it might not be doable with today’s classifications. An advanced level of acoustic data could be used, which contains data for specific drone models, in various flight modes, configurations, atmospheric conditions, different speeds and distances. The data would need to be measured consistently to maintain trust and quality in the database. If the data is absent or lacking, a theoretical level of acoustic data would be needed and generated to calculate of Noise Emission Profile.

#### 4.2.2. Drone Flight Profile

Drone flight profile data contains the drone’s information related to elevation, altitude, speed, and time of flight. This data gives the trajectory of the drone in a three-dimensional space. It can be used to predict the power usage, atmospheric state and similar parameters for noise calculation. It can also aid in forecasting and managing flight corridors. In air traffic management systems, radar is widely used to gather this data. However, utilizing radar will most likely be not feasible in a UTM due to the minimal radar cross-section of drones and the low operational altitude. A source of operation data could be through a communication platform where each drone transmits it’s flight data recordings to a database.

#### 4.2.3. Drone Operation

Drone operational data contains data related to the situational parameters of the drone. The data which is collected and managed, could be operational weight, procedure, atmospheric state, current speed and trajectory during operation and etc. Another type of data, such as change of drone geometry, might also be relevant (e.g. transportation of cargo or other modifications).

#### 4.2.4. Environment Data

The environment data consists of the surrounding conditions in approximation to the drone’s location or area that it operates in. The purpose of the environment database is to transform complex details into manageable parameters for the noise engine. The database will include geography, as it alters the sound through the topography, type of terrain and constructions in the area. The weather data is also relevant for calculations, as it contains temperature, humidity, rain, snow, wind power and wind direction. The primary purpose of weather data is to measure how it affects the sound. However, it can also affect flight paths and dynamics in the environment which is relevant to forecasting of the noise.

#### 4.2.5. Data and Database Quality

A challenge with the databases might be managing and ensuring a good enough quality data for the noise modelling service to achieve sufficient results. The model relies heavily on the quality of the input data and the different databases’ accuracy. The raw data cleaning process is essential to handle incorrect and corrupt data, to remove duplication and to fill in incomplete data within the data set. The data should be attainable in a short time, as the noise modelling should be quick enough to be used in a live U-space environment. The quantity of data and the amount of needed computer processing power also effect the result rate. Therefore, real-time calculations are a concern related to the operation of a noise modelling service (NMS). The resources, required to operate the NMS, should be aligned with the performance requirements set by the U-space. Comprehensive databases for aircraft models have been growing over many years. We would assume that such data and databases would increase for drones, resulting in more reliable, usable and faster data. Over the time, the NMS should learn what data is critical, sensitive or non-essential for achieving good results.

### 4.3. Noise Modelling Service

In this section, we present the details of the processes inside the noise modelling service and explain each component in detail. The aim for calculating noise is to end up with a $L_{E}$ or $L_{max}$. Here, $L_{E}$ is the sound exposure level given in a logarithmic measurement of the sound exposure over a time interval or event of a duration. $L_{max}$ is the maximum level of noise within a given time period. The calculation process of the NMS is shown in Figure 4. In a simplified term, NMS has a pre-processing module which gathers and manages the data, the noise emission profile which calculates the sound output, and the noise engine that performs the calculation of the noise descriptor.

#### 4.3.1. Pre-Processing

Pre-processing is a sub-process that manages and integrates the drone database, drone flight profile, drone operation, and environment data. It produces key parameters from the integrated data, which can be utilized later by other two modules i.e., the noise emission calculation profile and the noise engine. When the relevant data is unknown, one should consider the worst conditions within the drone class (e.g. maximum take-off mass). The acoustic and performance data is sourced from the drone database. The drone’s characteristics, such as operational weight, procedure and atmospheric state, are acquired and managed from the drone operation. The information of the drone in a three-dimensional space, to achieve height, speed, time and mission, is sourced from the drone flight profile. The surrounding conditions around the drone’s location, such as weather and geographical information, is collected and managed from the environment data.

#### 4.3.2. Noise Emission Profile

The noise emission profile process is used to calculate the sound pressure of the drone by considering its characteristics such as acoustics, operation and profile. The output is a sound pressure level at a specified distance, typically one meter, which can be used in a simulated or real scenario. Theoretical noise emission profiles for drone models could be stored and reused later in U-space or other applications.

#### 4.3.3. Receiver Point

Distance, geometry, angle and atmospheric properties influence the sound propagation path. This step aims to identify the receiver’s position to know the sound travelling distance. The receiver is the location where one wants to measure the noise level from the source, in this case, the drone.

#### 4.3.4. Free-Air Attenuation

The purpose of free-air attenuation is to give input on the rate of reduction in noise over distance. Two primary factors influence this rate and these are the distance and the atmospheric absorption. The atmospheric conditions vary between source and receiver. However, it might be practically irrelevant to include this variation. In a drone NMS, it is a short noise travelling distance, it might have a negligible impact on the final results, and there is a challenge in collecting such detailed data.

#### 4.3.5. Lateral Directivity

Most drones can fly in any direction, and with this, tilting of the rotors may apply. Lateral directivity handles the main propagation direction from the source.

#### 4.3.6. Lateral Attenuation

Lateral attenuation component considers the angle and distance to the ground and surrounding surfaces, and calculates the loss of energy due to distance and atmospheric absorption. Reflection and interference largely influence the results. Wind and temperature variations influence as well, but it may also increase the complexity of the calculation.

#### 4.3.7. Receiver Factors

The altitude of a drone is relatively low. Thus, the surrounding ground, constructions, topography and terrain must be considered during simulations. The lateral attenuation results have a significant relation with the receiver factors.

#### 4.3.8. Noise Scale and Metric

The output from the noise model is the noise scale and noise metric. The noise metric has two main types: Single event and total noise experience over time, depending on the desired outcome. For a single event, one must calculate $L_{max}$, which is based on the propagation path in a single straight line from source to receiver. Usually, the closest distance is chosen, as it has the highest probability of resulting the highest noise value. If it is desired to have the total sound energy in a specific event, e.g., a drones flight path from point A to B to C, one must calculate $L_{E}$. To calculate the noise over time, it is needed to add an iterative loop, which includes longitudinal directivity and integration along the flight path. The noise scale is presented in A-weighted, perceived loudness on frequency. This is a universally accepted standard for simulating the response of the human ear.

#### 4.3.9. Longitudinal Directivity

Longitudinal directivity calculates the effect of sound on a moving object in a longitudinal direction. Most importantly, during curved flight paths. Primarily it is the timing of the event that matters most during a flight path and, to a lesser degree, the longitudinal direction.

#### 4.3.10. Integration along the Flight Path

The Integration Along the flight path component does the computing by summing each part of a flight event. There are two suggested models: simulation models, and integrated models. Simulation models are the most time consuming of the two, as these models log received noise level over time for each new position. Integrated models calculate the $L_{E}$ as the drone flies in a straight path, while the turning effects are added adjustments.

### 4.4. Post-Process

We include the post-process component in the conceptual framework. It strongly connects to the noise modelling service and is relevant to the U-space system of systems perspective. The post-process may generate a noise contour or perform further analysis by using the delivered noise level from NMS and by forwarding to U-space, as seen in Figure 3. The post-process component can be highly complex, and a separate paper should be written on this. However, in this section, we present our main findings on the topic.

The post-process is assumed to be used to forecast or monitor an environment with multiple entities that coexist, focusing on the noise contour. The input data are highly unique for each scenario and are based on factors such as location, regulations, traffic, events, type of operation, and climate. As multiple drones might fly in the same area over a short period, the noise impact will increase. The post-process should perform grid operations by merging, subtracting, and adding these noise exposures to achieve a noise contour.

With the growth in drone traffic close to the general population, the increased noise in the environment might be annoying to people. The post-process could be used to analyse noise emission co-related with the annoyance in the area. If an area reaches its noise level limit, the U-space could manage the drones’ flight path. However, how the noise affects the environment is dependent on how its inhabitants experience the sound as noise. Studies show that noise and annoyance differ around the world. Hong Kong is more acceptable to high road traffic noise than Stockholm [62]. Noise acceptance level will differ not only for each area but also for the time of day and year. This indicates that a global rule for noise acceptance is difficult to achieve, or it has to be very conservative at best. Therefore, each given noise acceptance level needs to be nationally, regionally or locally driven.

We suggest a more suitable way of noise acceptance level which is to be considered on the basis of whatever is recommended for health reasons. World Health Organization has developed Environmental Noise Guidelines for the European Region [62]. In that report, recommendations are given to road traffic noise, railway noise, aircraft noise, and leisure noise. Drone noise does not have a recommendation and does not fall into either of the existing categories. Torijaa et al. [43] have studied the annoyance effect of drones in different environments, but more studies on health effects and annoyance of drone noise need to be performed. For the future, we recommend that U-space takes a basis in WHO recommended noise level when calculating flight corridors for the drones.

### 4.5. Noise Modelling Service Verification with a Use Case Scenario

As the noise modelling service is in a conceptual phase and has not yet been implemented in real life, we present a use-case scenario. The purpose of this subsection is to verify and validate the concept by describing how the proposed noise modelling service can be used in practice after it is implemented.

The case studies should have a set of representative drones from class C0–C4, and their performance data and acoustic properties should be stored in the drone database. Then a series of test flights should be performed where the calculated noise is compared with actual recordings. If possible, the first drone should be flown indoors to eliminate the complexity of an outdoor environment. The altitude and lateral distance to the recording point should be varied to test out the free-air attenuation and lateral attenuation algorithms. Kennedy, Garruccio and Cussen gives a good description of such a test regime with different altitudes in their recent paper [9].

#### A Proof-of-Concept: Use Case Scenario

To test the applicability and the validity of the proposed framework, we consider a use-case scenario that is described as follows: In order to validate the framework, a possible test arena for autonomous traffic could be Kongsberg Test Arena in Norway. In this test arena, different drones could be flown at different altitudes and routes. The drones flying in the area would have their acoustic and performance data collected and stored in a database. The drones could be equipped with tracking devices that report their flight paths to a tracking service, recording the drone flight profiles. The altitudes could for example be 30, 60, 90 and 120 m, since 120 m (400 feet) is the upper limit for drones in the open category. The routes should include flights over open areas and areas with many buildings, and with different distances to sound recording equipment placed at different sites throughout the area. After collecting the data related to drones, flight profile and operations, it is pre-processed in the NMS and the prepossessed data is then used for the noise emission calculation and the sound level. The result of the noise emission profile is then given to the noise modelling service to calculate maximum and average noise exposure for points or areas of interest, and these calculations should be compared against the recordings. The results from NMS is then delivered to U-space architecture.

In a theoretical setup, we want to calculate $L_{max}$ for a specific receiver point P in an open park. One drone, a DJI Phantom 4 Pro 2.0 directly overflies this point during the given time of interest. We assume that the data in the drone database says that this is a C2 class drone with a mass of 1.4 kg, a maximum speed of 20 m/s and has a sound pressure level of 76.9 dB(A) at a one-meter distance [63]. The drone flight profile tells the pre-processor the flight path: From point A to B, cruising at 30 m altitude the whole way. The drone operation database tells the pre-processor that no extra equipment or cargo was carried, leaving the sound level unaltered. The environment data tells the pre-processor that the weather was clear and still. The noise emission profile process then takes these parameters into account and can conclude that the sound pressure level for the drone is 76.9 dB(A) during flight.

The noise engine in the noise modelling service then calculates the the sound pressure for point P. Since the drone’s flight path is directly overhead lateral directivity and lateral attenuation are not taken into account, only free air attenuation. The inverse square law for sound attenuation over distance (for a point source) is [59]:(1)$$L_{2}-L_{1}=20\log\frac{d_{2}}{d_{1}}$$
where:

$L_{1}$ = Known sound pressure level at a reference location (76.9 dB(A) in our example)

$L_{2}$ = Unknown sound pressure level at a second location (receiver point in our example)

$d_{1}$ = Distance from the source to location of known sound pressure level (1 m)

$d_{2}$ = Distance from source to the second location (30 m)

Using this Formula (Equation 1), we can calculate that the noise exposure $L_{max}$ is 47.4 dB(A) for the point in question.

## 5. Conclusions and Future Work

Drones have an increased impact on our lives, both positive and negative. The concern is noise and the annoyance that drones inflict on communities. To better understand drones and their noise impact on the environment, we investigated what elements are important and to be considered for modelling drone noise and based on the investigation, our findings suggest that a noise modelling service for drones in U-space architecture does not exist and needs to be designed. In light of this gap, we proposed a conceptual framework for such a noise modelling service, with knowledge and inspiration from existing models established in the aviation industry. As aircraft and drones’ characteristics somewhat differ, necessary modifications were made. The noise modelling service consists of three main parts. The pre-processor that does the data management, the noise emission profile which calculates the noise from the particular drone, and the noise engine, which calculates the sound level based on the drone flying in an environment. To complete a noise calculation, there is a need for various data. Drone data is needed, such as acoustic and performance. Environmental data should be included, such as atmospheric state and topography. An important but challenging task would be to gather the necessary quality data to give a trustworthy result. The paper presents a theoretical use case with calculations for verifying the procedure of the framework. From this noise modelling service, we believe that regulators, researchers, operators and the communities can benefit by creating several scenarios to model the noise. It could also help to understand drone noise in airspace where humans are affected, both annoyance and health-wise, and take measures needed to reduce the noise impact on the community. Therefore, a noise modelling service could be a key part of a UTM and something both government and citizens would benefit from.

As a future work, we will develop and test the proposed noise modelling service in applicable settings to validate the concept and integrate the noise modelling framework with the U-space architecture.

## Figures and Tables

**Figure 1 ijerph-19-00223-f001:**
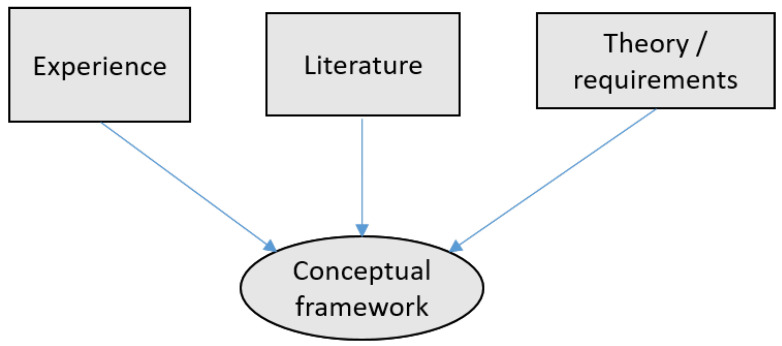
The sources for development of the conceptual framework [51].

**Figure 2 ijerph-19-00223-f002:**
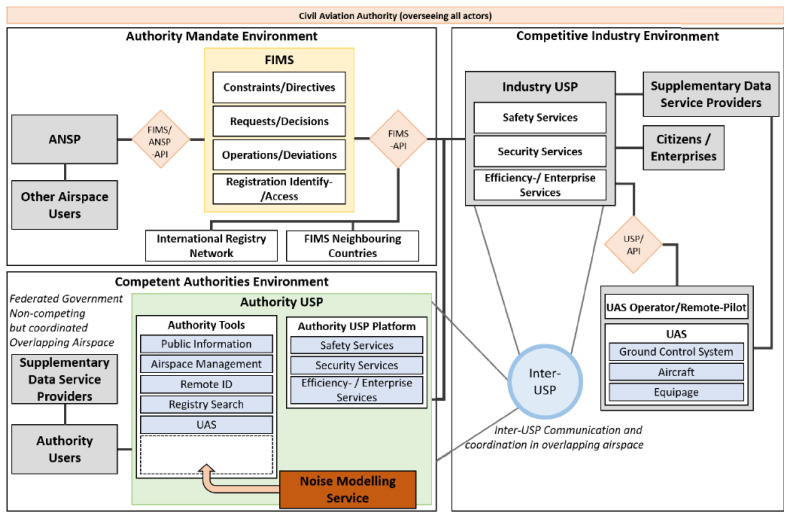
The U-space architecture for drone operations with the addition of a noise modelling service. Evolved from the Swiss definition of U-space [15].

**Figure 3 ijerph-19-00223-f003:**
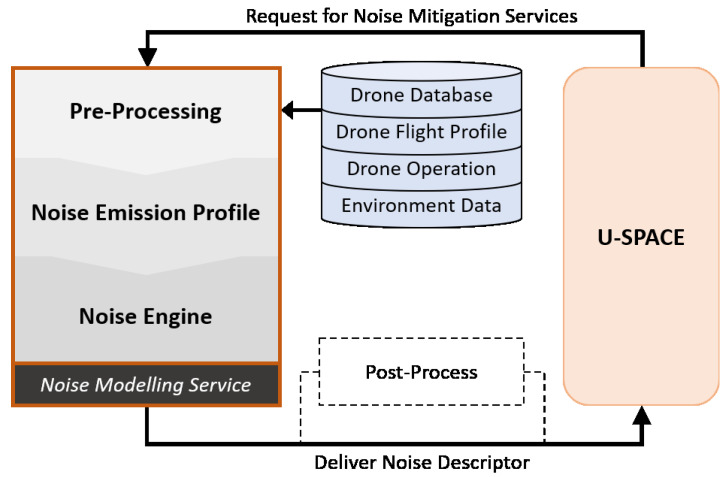
A Conceptual Framework for Drone Noise Modelling Service.

**Figure 4 ijerph-19-00223-f004:**
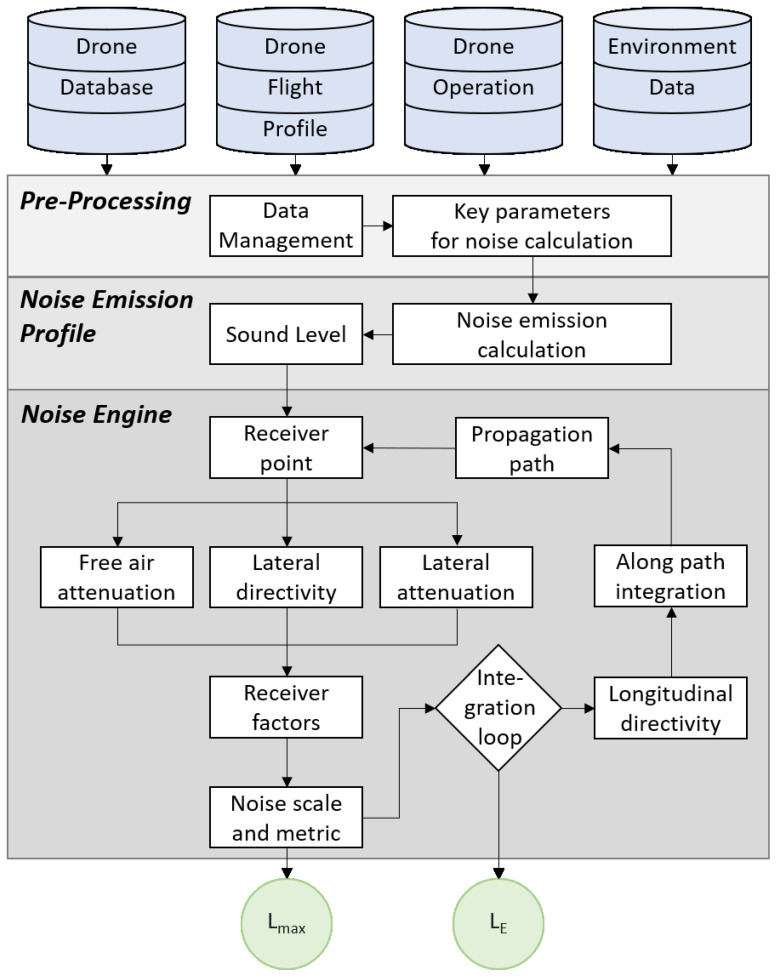
Functional block diagram for the Noise modelling service.

**Table 1 ijerph-19-00223-t001:** EASA classification of drone models, with noise levels found in the literature.

Class	Description	MTOM	Max Noise Level (EU)	Measured Min-Max
C0	Toy drone	<0.25 kg		60 dB [57]
C1	Hobby drone	<0.9 kg	85 dB [54]	40–87 dB [57,58]
C2	Prosumer	<4 kg	85 + 18.5 × $\log\frac{\mathrm{MTOM}}{0.9}$ [54]	40–70 dB [57,58,59,60]
C3	Professional	<25 kg		66–79 dB [61]
C4	Aero model	<25 kg		66–79 dB [61]

## Data Availability

Not applicable.

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
