# Peer review of "A Conceptual Framework Proposal for a Noise Modelling Service for Drones in U-Space Architecture"

_ijerph, 2021, doi:10.3390/ijerph19010223_

Round 1
Reviewer 1 Report
Section 2 must be improved. In the methods section you should list the methodologies you used to perform your work, which you say is the creation of a framework. Unfortunately, the authors list a series of preventive activities that the users have certainly followed but which add nothing to the work they propose. In this section the authors only need to describe the technologies used to develop the framework.
Section 3 must be improved. Move this section before Materials and Methods section.
Section 4 must be improved. There are not results and discussion showed in the first part of the section. According to what is reported in the outline of your work, in this section you should have reported the results of your framework. Actually, in the first part of the section you describe how you developed your framework, so you should move this section to the Materials and Methods section. A detailed description of the case study to be used to demonstrate the effectiveness of the proposed framework is missing.
116-118) The experience of the authors of each publication is clear there is no need to highlight it in the paper. If you wish to underline it, it is possible to add citations to other works that the authors have published and which certify their experience. Remove this section.
119-146) Bibliographic research activities are the basis of any scientific research, it seems superfluous to list these activities, also indicating the databases you have consulted and the keywords used. Remove this section.
147) Remove the “Theory / requirements:” subtitle . Start this section with “After reviewing the literature”…
223-230) This is the crucial part of this section. You should describe in detail how you developed the framework.
251) Do not use abbreviation such as i.e. I have seen that you often use this abbreviation, so I will not repeat this advice again, it also applies to the other occurrences.
399-657) Move this section in Materials and Methods section
417) Introduce adequately the noise modelling service topic.
Author Response
Please see the attached file for our explanation of the reviews.

Reviewer 2 Report
The paper perfectly fits into a recent field of the scientific literature that is still poor and deserving attention. The document is well written and covering all the aspect that it needs. Thus, my review will be short and only suggesting a couple minor aspects to be fixed before the paper’s publication.
In the introduction the authors correctly gave space to the important and impacting noise sources. I kindly ask them to add a bit more reference for those source, maybe 1-2 for each. I just want to suggest one for the road traffic noise, which is Bianco, Francesco, et al. "Stabilization of a pu sensor mounted on a vehicle for measuring the acoustic impedance of road surfaces." Sensors 20.5 (2020): 1239
In Chapter 3, when mentioning the paper analyzed, please recall the works with the author’s name. E.g. for line 246: “Jeon, S., et al. [37].”. This is a bit more stylish than reporting “numbers”, which are also not in order.
Avoid subchapters in the conclusions. Just produce a single, longer and comprehensive chapter.
Author Response
Please see our comments to the reviews in the attached file.

Reviewer 3 Report
The paper presents a critical topic that is of great interest to aerospace industry today. That is, drone noise reduction has not been overviewed, and few studies clearly addressed relevant problems that are in principle different from aircraft noise. The reviewer believes that the paper has good quality and can be published.
Author Response
Thank you.

Round 2
Reviewer 1 Report
Unfortunately, the authors did not highlight the changes made to the paper, making the reviewer's job more difficult. The paper is purely theoretical but the changes made by the authors have improved the presentation of the contents. Removing some unnecessary information made the document more readable.
It remains to improve the quality of the diagrams that contain some writings that appear blurry with some cut writings (Figures 1 and 2), moreover it is necessary to check the dimensions of the text which in some cases are undersized and in the printed version could be problematic.
Author Response
Hi,
Thank you for the review. Please see the attachment which contains our answers to the given review.

This manuscript is a resubmission of an earlier submission. The following is a list of the peer review reports and author responses from that submission.
Round 1
Reviewer 1 Report
In this paper the authors propose a conceptual framework for such a noise modelling service for drones. The authors also state that the framework is conceptualized based on noise modelling from the aviation sector. A noise modelling service can be used to model the noise to understand and take measures that are needed to reduce the noise impact on the community and to understand the accepted drone noise levels in different scenarios. The idea of this work is interesting, but the authors have to work on the presentation. First of all, I think they should remove the reference to the review from the title. In this work there is no review on the subject as only a few works are reviewed. A review article should cover at least 100 papers. The part of reviews they do is typical of any self-respecting scientific work. Authors need to focus on the metodology they intend to develop. On this, however, they need to be more detailed. Finally, they must develop a case study that can validate the procedure.
Section 2 must be improved. In the methods section you should list the methodologies you used to perform your work, which you say is the creation of a framework. Why do you start with the goals of your work? The objectives of your work must be clear to the reader by the end of the introduction, so you should move the objectives to the previous section or rename the section. Bibliographic research activities are the basis of any scientific research, it seems superfluous to list these activities, also indicating the databases you have consulted. From what you write in this section, your goal is to review the methods available in the literature and choose the one you think is most appropriate. So, yours is a review job? Instead, in the abstract you write that you have developed a new method. You should make this clearer. At the end of reading this section I think the whole section is superfluous. It would be enough to condense this activity in the introduction and eliminate the section altogether.
Section 3 must be improved. Section 3 lists some studies that have dealt with the noise produced by drones. This list is only partial as there are so many other works that have dealt with the problem. Some have studied the noise sources of drones, types of propellers and types of cart. Others have dealt with the problem of identifying drones in open and closed spaces with the use of algorithms based on neural networks.
Section 5 must be improved. This section is crucial throughout the paper. Here you present the methodology dealt with in this work. The idea is very interesting, the weakness is due to the fact that it is a theoretical treatment only. There is no case study that brings out the criticalities of the method, a practical case would be needed to give depth to the methodology. Use any drone, prepare the data for this drone and evaluate the exposure of the receivers on a flight path of your choice. This case study can validate your procedure. Furthermore, the part relating to the noise simulation model must be completely revised. You have to review this section as you first talk about exposure and then consider the attenuation of the noise in the open field. Furthermore, you should add specific definitions of the quantities and equations that allow you to calculate their values.
103-128) But these activities are the basis of all scientific research, it seems superfluous to list these activities, also indicating the database you have consulted.
129-142) From what you write in these lines, your goal is to review the methods available in the literature and choose the one you think is most appropriate. So yours is a review job? Instead, in the abstract you write that you have developed a new method. You should make this clearer.
147-153) Among the activities you have listed you have forgotten to list the security issues that are of crucial topicality. What do you think of people's safety? What happens if the drones fall on people due to a failt? What happens if drones are used for terrorist purposes? In this regard, you could add reference to the following papers: “UAV blade fault diagnosis”, “unmanned aerial vehicle inside closed environments”, "Sound Event Detection for Smart city safety", and “unmanned aerial vehicle detection in indoor scenarios”.
226-227) “none of the above studies worked on noise modelling.” It seems to me a difficult statement to share, instead there are several researchers who have created noise simulation models that can be applied to this case.
268) Do you have permission to publish this Figure?
286)UTM - Do not use acronyms until you have presented the full definition
295) Do not use abbreviation such as i.e.
315-321) To do a job well done, every drone on the market must be cataloged. This is an important job but it only needs to be done once and then you can add the new models.
331-332) Add references to support these statements.
337-340) This solution is more appropriate.
349) Table 1 should indicate the minimum and maximum noise values for all classes as it depends on the power that is delivered. The higher the speed of the propellers the greater the noise.
397-471) In this section, you explain how the noise simulation model works. You have to review this section as you first talk about exposure and then consider the attenuation of the noise in the open field. Furthermore, you should add specific definitions of the quantities and equations that allow you to calculate their values.
491-492) Now introduce the perception of noise, I think it is not appropriate to do so. Stay with the concept of exposure which still represents an essential measure to quantify noise disturbance.
Reviewer 2 Report
All the procedure is not clear. Authors spent many pages describing the procedure, without reporting any results. I suggest them to add them. If the authors believe to use these data for a consequent paper, this will be fine, but the present paper should them be a proper review, with more focus on just that. Otherwise, the present way is just an incoherent mixture. Thus, the authors should add some more results, or thinking about 2 different paper, a review and a methodology + result one.
Line 72. Please define for which noise source is intended the threshold. In fact, it surely depends on the type of sources, as also mentioned by Fredianelli, L., Carpita, S., & Licitra, G. (2019). A procedure for deriving wind turbine noise limits by taking into account annoyance. Science of the total environment, 648, 728-736.
It is now may 2021, and restricting the study to November 2020 is a bit too short. Please update the literature review to nowadays.
How are noise value evaluated? Please report it.
Please also add some picture for each category, as to help reader to have a visual correspondence.
Reviewer 3 Report
The paper has a very good quality, which would bring sound inspiration to the community. The ideas, methods and literature survey are clearly presented to guide/attract the reader with in-depth thinking.
The only point needing improvement is that a short section can be added to present a few demonstration cases regarding the verification and validation of the proposed framework. The cases need not be complicated but be simple and representative.
Round 2
Reviewer 1 Report
The authors did not handle the reviewer's comments and suggestions with due care. The new version of the paper is even more confusing than the first version. As suggested by the IJERPH journal template, a paper should contain the following sections: Introduction, Materials and Methods, Results and Discussion, Conclusions. By following this approach, the authors' work would have been much clearer to readers. Unfortunately, this was not the case. The authors have corrected the structure of the paper, now the proposal of a framework is highlighted, but unfortunately the framework was not presented in a satisfactory way.
The article must be completely revised. In section 2 the authors explain in detail a research methodology of another author [35] already adequately explained by the same. All this explanation does not add content to the paper and is not used to present the framework. Instead, it would have been necessary to explain in this section the framework that the authors developed.
Section 3 proposes an analysis of the literature on drone noise modeling: Important works are missing and in addition it should be moved before the presentation of the framework.
Section 4 introducing U-space architecture should be inserted in the section where the methods are presented. Similar thing for section 5.
Section 6 describes a use-case scenario and verification methods. This section needs to be completely revised. We need to present results and not yet formulas like the case of equation (1).
Finally, I have not found a section in which the answers to the three questions formulated in lines 90-94 are summarized. Are you sure you have fully answered these three questions? These are very challenging questions.
I am noticing that all the paper has significant formatting issues, this also blows the line numbers making the review more difficult.
Reviewer 2 Report
The authors improved a lot the paper, changing many pivotal aspects. My judgment is now positive, but i must ask the authors to read my previous round of suggestions and fullfil all the requests.
